# The Effects of Group Art Therapy on the Primary Family Caregivers of Hospitalized Patients with Brain Injuries in South Korea

**DOI:** 10.3390/ijerph18095000

**Published:** 2021-05-09

**Authors:** Nayoung Kim, Shin-Jeong Kim, Geum-Hee Jeong, Younjae Oh, Heejung Jang, Aee-Lee Kim

**Affiliations:** 1RN, Hallym University Chuncheon Scared Heart Hospital, Chuncheon 24252, Korea; supia1976@daum.net; 2School of Nursing, Hallym University, Chuncheon 24252, Korea; ghjeong@hallym.ac.kr (G.-H.J.); okim1108@hallym.ac.kr (Y.O.); hjjang@hallym.ac.kr (H.J.); 3College of Nursing, Sungshin Women’s University, Seoul 02844, Korea; aleekim@sungshin.ac.kr

**Keywords:** art therapy, brain injuries, caregivers, depression, self-efficacy

## Abstract

This study examined the effects of group art therapy on depression, burden, and self-efficacy in primary family caregivers of patients with brain injuries. This was a quasi-experimental, nonequivalent control group and a pre- and post-test design. This study was carried out in one national rehabilitation hospital targeting 41 primary family caregivers of patients with brain injuries. Group art therapy intervention was carried out three days per week comprising 12 sessions over four consecutive weeks. The experimental group (*n* = 20) received group art therapy, whereas the control group (*n* = 21) did not. We used a time difference method to minimize the risk of contaminating the control group by sampling sequentially. For depression, although there was a significant difference after the intervention (*t* = 3.296, *p* = 0.004), the mean difference score was not statistically significant between the experimental group and the control group (*t* = 0.861, *p* = 0.395). The experimental group showed a significantly greater decrease in burden (*t* = 2.462, *p* = 0.020) and significantly greater improvement in self-efficacy (*t* = −6.270, *p* < 0.001) than the control group. Group art therapy may be an effective nursing intervention for primary family caregivers of patients with brain injuries.

## 1. Introduction

### 1.1. Patients with Brain Injury and Their Families

Brain injury refers to a state in which the brain cannot function due to an abnormality in the nerve tissue from internal or external causes [1]. It is also defined as a physical disability such as gait disorder or disability of daily life movements caused by a brain lesion [2]. In addition, it results in varying degrees of physical and psychological health problems up to death [3]. The incidence of brain injuries differs worldwide from 811 per 100,000 in New Zealand to 7.3 per 100,000 in Western Europe [3]. In South Korea, brain injury was ranked as the third leading cause of death following cancer and heart disease during the past decade, with an incidence of 45.8 per 100,000 [4].

Brain injury is characterized by movement disorders, paralysis, speech disturbance, and sensory and cognitive impairments depending on the location of the brain lesion, requiring long-term care and often requiring help in all areas of daily life from family members [1]. These characteristics of brain injury may be stressful for patients’ family caregivers. In Korea, patients should have caregivers to support them and remain nearby during their hospital stay, even if admitted to a professional medical institution. The primary caregivers remain with the patient almost daily and even overnight. This practice is in line with the prevailing view of the Korean people that they are responsible for and should be involved in patient care as much as possible [5,6]—presenting families with the difficulty of long-term care.

Lack of information about their family member’s diseases and uncertainty about the future, handling of the disease itself, or sleep interruptions may also aggravate the primary caregivers’ health problems [5,7]. In addition to these difficulties, opacity in predicting the prognosis of the disease process, which requires long-term observation and care, not only affects the individual patient but also creates a burden for the family caregivers at the patient’s bedside [3].

Family burden is the subjective negative reaction of family members to the stressful situation of caring for the patient [6,8]. Patients are increasingly reliant on their families, whose burden increases because of the difficulty of short-term recovery. If caregivers’ burdens are not relieved or become overwhelming, life becomes unstable and eventually entails health problems that may directly affect the patient under their care [5,6]. Thus, the burden of caregiving may negatively impact the entire family’s functioning [8].

Depression, which affects the patient and the caregiver, has been reported as the most frequent psychological symptom experienced by long-term caregivers [7]. As the burden of care increases, caregivers are more likely to experience anxiety and depression [9].

Self-efficacy is defined as a belief that enables the individual to successfully perform the actions needed to obtain results [10], leading them to cope better with difficulty and frustration by concentrating on what they can do well. Thus, it is not a way to avoid reality; instead, it is a positive method for overcoming difficulty [11,12,13]. Self-efficacy theory, based on Bandura’s social cognitive theory [14], is an important concept for human mental health that may improve the quality of life, help overcome human suffering, and further suggest the possibility of new hope [13].

Previous studies found that more than half of caregivers had depressive symptoms, they had a heavy burden of caring, and the nursing needs of patients with brain injury were higher than those for other chronic diseases, negatively impacting the well-being of the primary family caregivers [5,6,7,8]. These results indicate that intervention methods are needed to minimize the burden and depression of patients’ family members. Positive changes resulting from psychological programs may influence patient care by promoting the psychological health of the primary family caregivers who care for them. However, most of the research conducted in Korea reporting difficulties of the primary family caregivers of patients with chronic diseases such as brain injury have been survey studies [5,6,7,8,15].

### 1.2. Benefits of Art Therapy

Expression in art may be regarded as an expression of self-image [16,17]. Art therapy (AT), a technique for people to resolve their internal conflicts and satisfy their desires, is known not only for relieving stress but also for improving one’s sense of accomplishment and confidence [18]. Through AT, the individual expresses negative feelings of the self and sublimates them, thereby increasing positive emotions and self-growth [19]. AT may serve as a substitute therapy to sublimate psychological conflict, overcome frustration, and stimulate overlooked aspects of life beyond the limits of verbal communication [13,20]. Riley reported that when creativity is introduced into problem-solving, art can provide fresh viewpoints and excitement [21]. Research with adults, elderly stroke patients [22,23], caregivers of stroke patients [7], and adolescents experiencing family conflict [13] have shown that AT improves their quality of life by reducing depression and stress and increasing self-efficacy and self-esteem. Employing AT with parents of children with disabilities reduces depression and stress, increases a sense of parenting efficacy, provides psychological stability, and positively effects parental happiness [24].

Group art therapy (GAT) is a form of psychotherapy that combines group therapy and art therapy. Group members share their emotions through interactions within the group and induce insights by forming therapeutic relationships [12]. Three merits of GAT are that, first, it provides the personal experience of engagement with creative activities; second, it easily allows the expression of emotions using art materials; and third, it is especially useful for people who have difficulty in verbal self-expression because the work itself has a symbolic meaning [18]. Finally, as all group members can participate at the same time, it provides personal and collective experiences at the same time, thereby giving a variety of meanings to all group members [13]. Experiences are expressed more freely and easily than during rational psychotherapy. GAT may raise self-efficacy by allowing group members to solve common problems and affording them a chance to observe the results of their activities with mutual support.

In the Korean hospital setting, the caregivers, who are usually the patient’s family members, must be present at all times with the patient. Thus, psychosocial support services for caregivers may be beneficial for hospitalized patients. However, it is difficult to find studies that evaluate GAT’s effects on primary family caregivers, who are indispensable to the patients. Therefore, we employed GAT with the primary family caregivers of hospitalized patients with brain injuries to examine the GAT’s effects on depression, burden, and self-efficacy.

### 1.3. Purpose

The purpose of this study was to examine the effects of GAT on the (a) depression, (b) burden, and (c) self-efficacy of family members, who are the primary caregivers of patients with brain injuries in a community rehabilitation hospital.

## 2. Theoretical Framework

Based on Rubin’s cognitive-behavioral art therapy (CBT) [25], Bandura’s social cognitive theory [14] was used as the theoretical framework for this study (Figure 1). CBT suggested that art is a cognitive process and creating art based on thinking about, identifying, and understanding emotions that has profound impacts on human behavior [25]. This social cognitive theory is based on a triadic reciprocal causation: Personal, Environmental, and Behavioral [14].

In this study, Personal contains potential stressors such as depression, burden, and self-efficacy. Environmental means the aspect of the setting that influences an individual’s ability to complete a behavior successfully, and it may improve emotional, cognitive, or motivational processes. We tried to alter the conditions of the caregiver with GAT. Behavioral refers to the response an individual exhibits after performing a behavioral, emotional, cognitive, or motivational process. We hypothesized that if the caregiver experiences a positive outcome with successful learning as a result of performing GAT behavior, their depression and burden would be lessened, and self-efficacy will increase.

Using Bandura’s theory [14], we aimed to lessen in primary family caregivers (a) depression and (b) burden and increase (c) self-efficacy by improving emotional, cognitive, and motivational processes (Figure 1). We used this theory in our study because behavioral change is influenced by the interaction of three types of determinants: Personal, Environmental, and Behavioral. In addition, art is a cognitive process that profoundly impacts human behavior (Figure 1).

## 3. Materials and Methods

### 3.1. Design

This study employed a quasi-experimental, nonequivalent control group and a pre- and post-test design (Figure 2).

### 3.2. Setting and Participants

This study was carried out in one national rehabilitation hospital located in Chuncheon City, Gangwon-do Province, South Korea. A convenience sampling technique was employed to recruit primary family caregivers. The recruitment notice, posted on the hospital bulletin board, asked potential participants to directly contact the principal investigator (PI) if interested. The 42 participants were enrolled and assigned to the following two groups: (1) the experimental group (Exp.), which received GAT during the patients’ hospital stay, and (2) the control group (Con.), who did not receive GAT (Figure 3).

We used a time difference method to minimize the risk of contaminating the control group by sampling sequentially [26]. Following completion of the recruitment process for the control group, the GAT intervention for the experimental group began.

Power analysis for a two-group comparison with a *t*-test with a significance level of 0.05 was conducted using G*Power 3.1 [27]. A minimum of 32 participants would be required to detect between-group differences with 70% power and a medium effect size of 0.8. Of the 42 participants recruited, the data for one of the participants in the experimental group were excluded at post-test because the information on the variables was incomplete.

Criteria for the inclusion of participants were (1) primary family caregiver caring for a hospitalized patient with brain injury and (2) not receiving programs similar to the GAT. Participants were excluded if they (1) did not want to participate in the study, (2) had a past diagnosis of a depressive disorder, or (3) were taking medication for a psychological problem.

### 3.3. Measurements

#### 3.3.1. Depression

The standardized “CES-D” Korean version, revised by Chon and Rhee [28] to better fit the specifics of Korean society and culture was used to assess participant’s depression level. The original instrument was the Center for Epidemiological Studies Depression Scale (CES-D), developed by the American Mental Health Research Institute for the epidemiologic study of depressive symptoms within the general population. This self-report questionnaire consists of 20 items recording experiences during the past week using a 4-point Likert scale: 0 points (less than 1 day per week), 1 point (from 1 to 2 days per week), 2 points (3 to 4 days a week), and 3 points (more than 5 days per week). Higher scores indicate a higher level of depression. The Cronbach’s alpha coefficient was 0.91 in the original study and 0.85 in this study.

#### 3.3.2. Burden

The participant’s burden was measured using the burden tool developed by Suh and Oh (1993) based on previous research and literature review [15]. It consists of 25 items, each scored on a 5-point Likert scale ranging from 1 (strongly disagree) to 5 (strongly agree). A higher score indicates a greater burden on caregivers. Cronbach’s alpha coefficient was 0.89 in the original study and 0.84 in the current study. 

#### 3.3.3. Self-Efficacy

Self-efficacy was measured using a tool reconstructed by Cha [29] and revised by Han [11] based on that developed by Sherer et al. [30]. It consists of a questionnaire with 24 items using a 5-point Likert scale ranging from 1 (strongly disagree) to 5 (strongly agree). A higher score indicates a higher level of self-efficacy. The Cronbach’s alpha coefficient for the original study was 0.89 and 0.87 for this study.

#### 3.3.4. Content Validity

A seven-member expert panel conducted a content validity test to examine the suitability of all instruments used in this study. The number of members was selected based on the fact that the ideal number on an expert panel is approximately 10 individuals [31]. Of these, three were adult health nursing professors, two were nurses working in a hospital for more than 20 years, and two were art therapists. The CVI (content validity index), showed that all instruments exceeded 80% (depression: 92%; burden: 96%; and self-efficacy: 97%), which is considered acceptable.

### 3.4. Training of the Research Assistant

Before the start of this study, the authors oriented the research assistant regarding the study processes. The research assistant worked as a registered nurse for two years in a general hospital and has a bachelor’s degree. The training lasted about four hours, one hour daily for four days, and included measuring variables related to this study. The questionnaire was distributed and collected by the research assistant. In addition, the first author observed the research assistant to ensure that measurement was performed correctly throughout the study. During the GAT intervention, the trained research assistant helped prepare the GAT activities, stayed in each session, and guided the participants to raise their hands and ask questions at any time.

### 3.5. Group Art Therapy Intervention

We set goals to be accomplished considering the program process’s impact and results. In this study, the goals of the GAT phases were as follows:Group formation phase (Sessions 1–5): Builds trust and intimacy among the group members, reduces their burden, and adapts them to and induces interest in GAT.Ego search phase (Sessions 6–9): Explores and recognizes an individual’s inner feelings and promotes self-efficacy through interactions among group members. It also reduces stress and anxiety, promotes understanding of oneself and others, and facilitates group interaction.Self-acceptance phase (Sessions 10–11): Enables an individual’s recognition of the changes in themselves, makes them confident through emotional stability, and establishes a positive self-image.Closing phase (Session 12): Organizes and closes the process through the group activity of “Salt Mandala”.

GAT was carried out in a program room separated from other rooms and had an exhibition space that held the artworks that the participants created. This location was selected because a confidential and secure environment is needed when implementing the AT; the AT environment directly affects participant’s level of participation and expression activities [16].

At the time of the study, the experimental group received GAT for one hour per day for three days per week on alternating days (Monday, Wednesday, and Friday) between 2:00 and 3:00 p.m., for a total of 12 sessions over four consecutive weeks. This particular time was already set for rehabilitation exercises for the patients with the help of a physical therapist or volunteer in the rehabilitation treatment room; thus, it was a relatively free time for caregivers that would not affect patient care. Regarding the frequency of GAT, the effect of AT is considered more effective when applied about two times a week [13] and for 1–2 h each session [17]. The control group did not participate in any activities related to art. Each group had seven members, and membership was determined by the length of their hospitalization. Therefore, the experimental group consisted of three GAT teams, and group activities were organized by team.

The first author, a professional art therapist licensed in AT, conducted the program based on her clinical experience of more than 10 years. The program consulted with and was guided by two experts. The contents of the GAT are summarized in Table 1.

Each GAT session lasted one hour and contained (a) an introduction (10 min), (b) work time (30 min), and (c) wrap-up (20 min), including the presentation of the artwork to another participant and a discussion among group members.

Introduction: At the beginning of each GAT session, the researcher started a warm-up activity by telling a relevant story for each theme or introducing an activity, identifying goals, and checking the materials needed for the session.

Enhancement: The first author then presented the art materials and invited the participants to engage in the activity. The use of the medium of art was based on Rubin’s theory [25], and each program session was composed and planned according to the guidelines of Choi [16] and the Korean Art Therapy Association [17]. In addition, the first author created all art lesson plans and guided the participants in the use of the materials. The participants completed drawings in response to the instruction and were informed that they had 30 min to complete the tasks.

The program of each session consisted of art-making activities and group artworks with various types of art materials. The instructor presented and demonstrated how to use the materials and then allowed the participants to practice using the materials. The art materials used included pencils, crayons, colored pens, colored pencils, 4B pencils, watercolor, India ink, markers, pastels, oily clay, salt, magazines, erasers, glue, scissors, various sizes of drawing paper (A2, A3, A4 size), wire, and items that could be decorated. Orr [32] suggests that painting can help participants talk safely about difficult experiences because it can be easily controlled, and the use of watercolors is useful for improving emotional expression because of its fluid properties. The activities that were performed were mostly facilitated, and the instructor waited for the participants to complete the work.

Wrap-up: After the participants finished the creative work, they were asked to show their work to the audience. Thus, they had an opportunity to present their own work and discuss their experiences. Each participant’s artwork was digitally photographed using smartphones and stored for evaluation (Figure 4). During the discussion, they shared their artwork and their feelings. They were also encouraged, but not required, to discuss their feelings and memories related to their trauma history and difficulties. Participants were also permitted to ask questions to make sure they understood the instructions.

After completing the 12 sessions of GAT, each group member was given time to organize their own book and make front and back covers for it. Following this study’s completion, we provided group art therapy for the control group who wanted to participate in the program (*n* = 14) to provide equal opportunities for all participants.

### 3.6. Data Collection

Data were collected from 1 August 2017 to 15 December 2017. The study was presented personally to the staff members of the rehabilitation hospital to recruit participants. To obtain informed consent, all interested participants were interviewed and informed of their right to decline participation. They were informed that the data collected would not be treated individually but as a group.

The pre-test was conducted to examine the homogeneity of the sample characteristics and baseline depression, burden, and self-efficacy levels immediately before beginning the GAT. A post-test was conducted two weeks after completing the GAT intervention using the same questionnaire as in the pre-test.

### 3.7. Ethical Considerations

Before conducting this study, approval was granted by the Hallym University Institutional Review Board (HIRB-2017-031), to which one of the researchers is affiliated; issues of voluntary participation, anonymity, and confidentiality were addressed. The IRB confirmed that the study did not violate human rights and that all content and processes conformed to standard research ethics requirements.

### 3.8. Data Analysis

Analyses were conducted using SPSS version 21.0 (IBM Corp., Armonk, NY, USA). Descriptive statistics and Student’s *t*-test were used for data analysis. The level of significance was set at 0.05.

## 4. Results

### 4.1. Descriptive Results and Homogeneity Test of Demographic Characteristics and Outcome Variables

Table 2 shows the baseline demographic characteristics and outcome variables for both groups.

Of the 41 participants, 12 (29.3%) were male and 29 (70.7%) were female; the mean age was 55.8 (±12.4) years, with a range of 27 to 82. In addition, 27 (65.9%) reported religious affiliations, while 14 (34.1%) did not. Concerning their relationship to the patient, spouses made up the largest proportion, 19 (46.3%), followed by sons (29.3%), daughters (9.8%), and others (14.6%).

Regarding patient characteristics, 30 (73.2%) were male and 11 (26.8%) were female. The mean age of the patients was 61.6 ± 15.8 years, with a range of 24 to 89. Their diagnoses were cerebral infarction (56.1%), cerebral hemorrhage (31.7%), and brain injuries due to accidents (12.2%). Regarding the number of hospitalizations, including that of the current study, the most frequent was two times (34.1%), followed by three times (22.0%), the current one being the first time (17.1%), four times (2.4%), and 24.4% more than five times.

There were no significant differences between the two groups regarding any demographic characteristics of caregivers, such as gender, age, religiosity, relationship with the patient, and inpatients’ characteristics such as gender, age, medical diagnosis, or the number of hospitalizations. In addition, there were no significant differences between the two groups in the levels of depression, burden, and self-efficacy. Therefore, the two groups may be considered homogeneous.

### 4.2. Mean Differences in the Dependent Variables between the Two Groups

The mean difference (post-intervention minus pre-intervention) scores on depression, burden, and self-efficacy between the two groups are presented in Table 3.

The *t*-test found a significant difference between the experimental and control groups in the mean difference scores for burden (post- vs. pre-test scores) (*t* = 2.462, *p* = 0.020) and self-efficacy (*t* = −6.270, *p* < 0.001). However, the mean difference score for depression (*t* = 0.861, *p* = 0.395) was not statistically significant. 

## 5. Discussion

This study showed that GAT effectively decreased the burden and increased the self-efficacy of primary family caregivers of hospitalized patients with brain injuries. In the current study, the degree of burden in the experimental group was significantly decreased, and self-efficacy significantly increased compared to the control group. Although, the depression level of the experimental group did not decrease significantly after GAT compared to the control group, the score of the post-test was decreased significantly compared to that of the pre-test.

The different outcome of the significance of depression level is somewhat inconsistent with the results of other studies [22,33,34]. For adolescents experiencing family conflict [13], the depression level was significantly lower in the group receiving GAT than in the group that did not. In addition, the results of a study of 39 women with breast cancer showed clinically significant improvements on the scale of depression [35]. Regarding the current results for depression level, we speculate that there are two likely causes. One possible explanation is that the GAT intervention lasts for a short period time (4 weeks), while depression intrinsically takes a long time to change [7]. Another explanation is that there might have been some difficulties in eliciting the participants’ intrinsic feelings in front of the other group members during GAT interaction.

This study’s, results, in which burden is significantly reduced, confirm the study by Nobel et al. [36] of a patient with diabetes. One of the GAT intervention’s goals was to ease the emotional burden through creative artistic expression, which was achieved in this study. AT has been found to help people confront their realities and sublimate their negative emotions [37]. In our GAT intervention, the participants were given opportunities to explore, process, and reflect on the burden of caregiving. An analysis of the qualitative and quantitative data partially supports the hypothesis that an AT group within a school setting can help increase coping among eighth-grade students who are at risk of a poor transition to high school [20,38]. GAT may allow participants to express and reduce their negative emotions through their artwork. Monti et al. [33] reported that the social relationships developed through GAT could positively affect participants’ identity and disease-related experiences. Although this study targeted primary family caregivers who were not patients, its application for patients can be considered in the future.

In the current study, GAT was found to have a significant positive effect on the primary family caregiver’s self-efficacy, consistent with Cira and Michelle’s study [39] result that participants who had received GAT showed greater self-efficacy than those who had not. Children attending a GAT intervention program displayed significantly greater self-efficacy than children who did not [40]. GAT has also been found to positively impact the self-efficacy of working mothers [12]. Kim [13] implied that the self would mature through associations with pictures. Conceivably, observing one’s own completed artwork and listening to comments and compliments from others can strengthen self-efficacy within a group activity more than on the individual level, potentially leading the participants to experience fulfillment and accomplishment, which enhance the feeling of self-efficacy [12]. Concerning increased self-efficacy, the improved confidence created through completing the artwork could influence positive outcomes [13,41]. Moreover, during the wrap-up activities, such as presentation and discussion, self-efficacy can be facilitated by revealing oneself to others and receiving their support.

Arts-based interventions such as GAT are innovative and powerful resources that can be applied at a relatively low cost in a variety of settings in such a way that the participants find it easy to accept, and that serve to empower people who create artworks. It thus deserves serious attention as an intervention for caregivers [9,42]. Through GAT group dynamics and mutual support may contribute the positive results of this study. GAT’s method may allow participants to create an atmosphere of emotional well-being, experience new kinds of activities, and develop their creative abilities, thereby promoting their self-efficacy.

The findings of this study suggest that GAT may be one of the effective nursing interventions that can be employed with primary family caregivers. Furthermore, they indicate a positive impact and may provide empirical support for using GAT in a hospital setting and suggest that GAT may be used for other groups also. A small body of studies now exists documenting AT as a caring modality that has been measured and shown to yield statistically significant positive results in various populations at different ages and with a variety of difficulties. Moreover, GAT’s development, suited to a particular care setting, is required to enhance family-centered care in nursing practice. Further studies are also proposed to investigate whether other activities are needed to determine whether the effectiveness of GAT is comparable with that of other artistic activities and whether it produces better results.

### 5.1. Implications for Nursing Practice

This study shows that GAT is an intervention that may play a collaborative role in supporting the well-being of the family members of hospitalized patients with brain injuries. In recent years, family-centered care has been emphasized, especially in nursing practice. Nursing focuses on an optimal level of health for patients and their family members among the general population. In this respect, GAT, which has been shown to be effective in this study, can be used by family members who care for patients.

Our study’s results were taken to support GAT as an efficient tool for primary family caregivers who are required to participate in the patient’s rehabilitation and for the caregivers of patients with other types of chronic and long-term disease. Therefore, the results of the present study might be used by clinical health professionals as evidence-based data for providing family-centered care design programs to alleviate the difficulties of family caregivers.

Health professionals, including nurses, are in a unique position to judge whether GAT is likely to be beneficial for the caregiver of family members. In addition, nurses may encourage caregivers to participate in such activities. One of the dilemmas for implementing the intervention may be a shortage of time and funding. However, once a GAT program is established, it may get help from the hospitals for human resources and financial support. As a result of participation in GAT activities, nurses may indirectly recognize the benefits and effects of the program by sharing the participants’ thoughts and feelings and become more open to these kinds of activities that are helpful to the caregivers. Finally, it may contribute to family-focused nursing.

This study’s findings may also be used to assess and intervene to help primary family caregivers with difficulties such as depression, burden, and self-efficacy. GAT may also be applied as an intervention for patients and family caregivers who have family members with various types of chronic diseases. In addition, the differences in GAT effects, such as the background of participants and patients within the group could be considered. In conclusion, this result may contribute to the well-being of primary family caregivers who take care of patients with brain injuries.

### 5.2. Limitations

The findings of this quasi-experimental study should be considered in light of several limitations. First, a nonequivalent control group pre–post-test design was adopted instead of a randomized controlled design. Second, although the sample size in this study was acceptable, it was relatively small. Third, we did not account for mediating or moderating variables that may have affected the results. Finally, there was no follow-up to determine the effects of this study after a greater length of time.

It would be beneficial to address the study’s limitations considering the research design and recruit more participants from a wider range of locations and cultural backgrounds to see whether the results of GAT may be generalized. Moreover, to consider the mediating or moderating variables and to determine the longer-term effects of GAT participation, a follow-up study is recommended.

## 6. Conclusions

This study examined the effectiveness of a GAT intervention within a hospital setting targeting primary family caregivers of hospitalized patients with brain injuries. In this study, GAT decreased burden and enhanced the self-efficacy of the participants. The study results partially support the potential benefit of GAT to family-centered care of hospitalized patients with brain injuries and its extension to other chronic diseases in the future.

## Figures and Tables

**Figure 1 ijerph-18-05000-f001:**
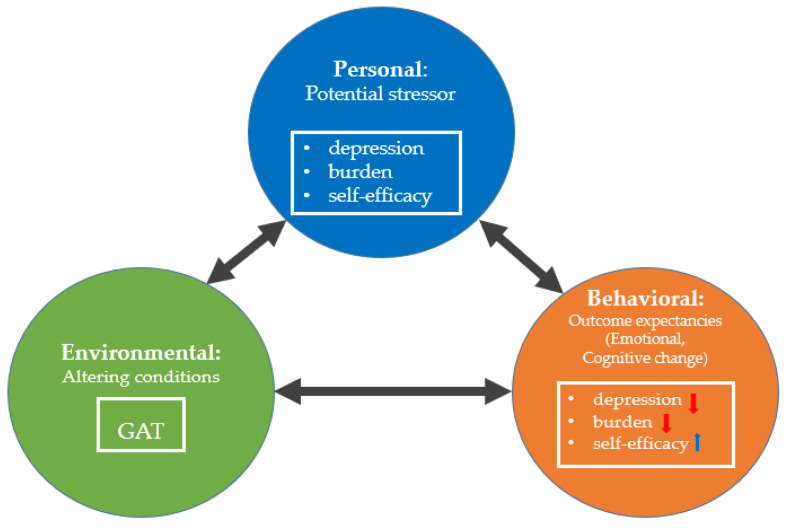
Theoretical framework. GAT refers to group art therapy.

**Figure 2 ijerph-18-05000-f002:**
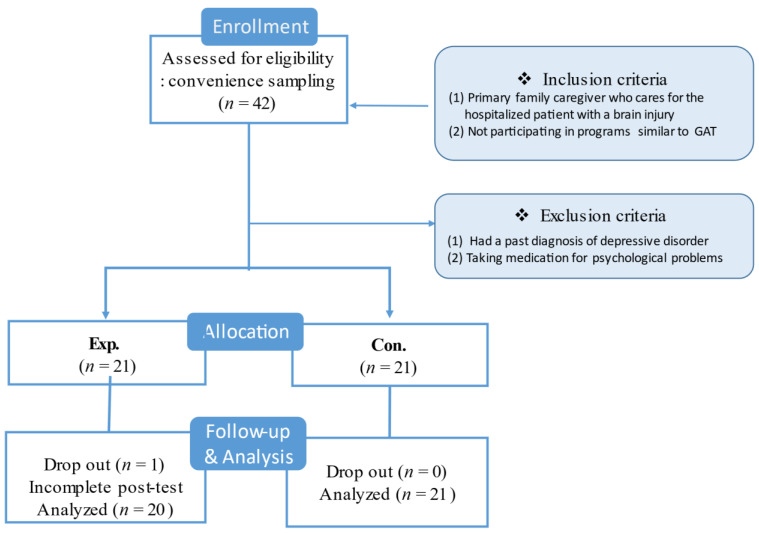
Study design. GAT refers to group art therapy.

**Figure 3 ijerph-18-05000-f003:**
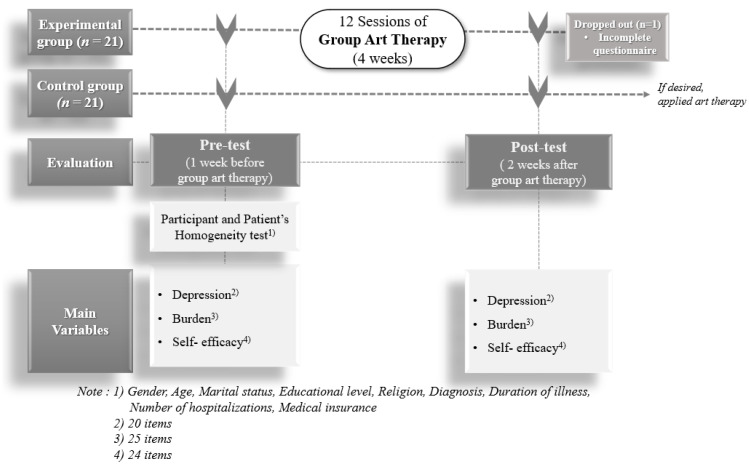
Study process.

**Figure 4 ijerph-18-05000-f004:**
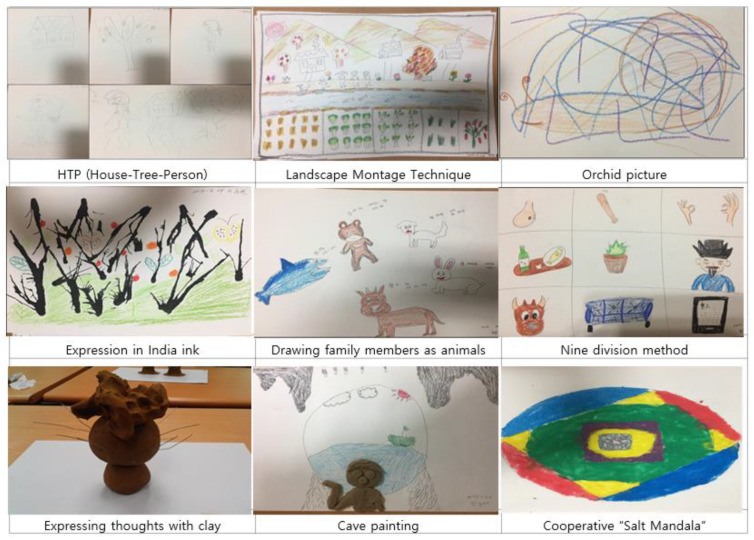
Artworks completed during the group art therapy intervention.

**Table 1 ijerph-18-05000-t001:** Contents of the group Art Therapy intervention

Session (Duration)	Theme	Contents	Method
I (1 h)	HTP (House-Tree-Person)	Introduction Self-introduction Motivation Identifying goals Introducing activities	Lecture PPT
		Activities Place the A4 paper horizontally and write your nickname Draw on the paper, a house, a tree, and a person doing something Explain precautions to avoid drawing people as cartoons	A4 paper Pencil Eraser
		Wrap-up Present their work Discussion about the participants’ artwork Q & A Notice for the next session	Presentation Discussion
II (1 h)	LMT (Landscape Montage Technique)	Introduction Identifying goals Introducing activities	PPT
		Activities Place the A3 paper horizontally and draw a border around the paper with a black marker Draw a landscape with ten elements such as a river, mountain, field or rice field, road, house, tree, people, flower, animals, stone, and so on Add other elements and color them in	A3 paper Black marker Colored pencil Crayon
		Wrap-up Present their work Discussion about the participants’ artwork Q & A Notice for the next session	Presentation Discussion
III (1 h)	Orchid picture	Introduction Identifying goals Introducing activities	PPT
		Activities Relieve tension by telling everyday stories to participants Let them express themselves freely using prepared art tools with lines rather than drawings Try to find the shape in the pictures Express the shape in the figure that they found	A3 paper Watercolor Colored pencil Marker pen
		Wrap-up Present their work Discussion about the participants’ artwork Q & A Notice for the next session	Presentation Discussion
IV (1 h)	Figure drawing from the frontal perspective	Introduction Identifying goals Introducing activities	PPT
		Activities Relieve tension by telling everyday stories to participants Order the group members When a person draws a picture on A2 sized paper, the next person looks at the picture thinking about the intention of the drawing and then draws the sequence, until the last person completes the work	A2 paper Crayon WatercolorColored pencil Marker pen
		Wrap-up Present the intentions of their drawings Discussion about the interpretation of others’ intentions Q & A Notice for the next session	Presentation Discussion
V (1 h)	Expression in India ink	Introduction Identifying goals Introducing activities	PPT
		Activities Drop the India ink on a piece of A2 paper using a brush Freely move the India ink that falls on paper by blowing it Think about the image that comes to mind when looking at the ink shapes Complete the painting by applying other materials over the India ink Think about the feeling of India ink	A3 paperIndia ink Colored Pencil Crayon Watercolor
		Wrap-up Present their work Discussion about the participants’ artwork Q & A Notice for the next session	Presentation Discussion
VI (1 h)	Drawing family members as animals	Introduction Identifying goals Introducing activities	PPT
		Activities Think about each member of your family and their personality, role, and position in the family Think about animals similar to your family members Draw family members as animals without limiting the position or size of the animals Label family members within the animal picture	A3 paper Crayon WatercolorColored pencil Marker
		Wrap-up Present their work(the reason for aligning certain family members with certain animals) Discussion about the participants’ artwork Q & A Notice for the next session	Presentation Discussion
VII (1 h)	9 division method	Introduction Identifying goals Introducing activities	PPT
		Activities Separate 9 spaces on an A4 paper Let feelings come up by thinking about the most thoughtful person Draw a picture with the thoughtful image pointing counterclockwise from the top right Complete the picture	A4 paper 4B pencil Eraser Crayon WatercolorColored pencil Marker pen
		Wrap-up Present their work Discussion about the participants’ artwork Q & A Notice for the next session	Presentation Discussion
VIII (1 h)	Expressing thoughts with clay	Introduction Identifying goals Introducing activities	PPT
		Activities Close your eyes Think of what occupied their thoughts recently when touching the oily clay Express their thoughts in clay	Oil-based clay Wire Decorative items
		Wrap-up Present their work (similar and different points with the clay) Discussion about the participants’ artwork Q & A Notice for the next session	Presentation Discussion
IX (1 h)	Cave painting	Introduction Identifying goals Introducing activities	PPT
		Activities Draw a circle in the middle of the A3 paper with pencil The researcher asks the participants to close their eyes and describe in detail a dark, invisible cave, the feeling of losing your way in the cave, the process of finding the cave entrance, and the feeling after finding the entrance Have participants imagine the process according to the researcher’s description Participants open their eyes and think about the process and display their feelings of the cave in the circle, and their feelings regarding the landscape outside the cave outside the circle Complete the picture Make their appearance on the painting in clay when they find the entrance of cave	Oil-based clay A3 paper Crayon Watercolor Colored pencil Marker
		Wrap-up Present their work Discussion about the participants’ artwork Q & A Notice for the next session	Presentation Discussion
X (1 h)	Self-portrait &others’ portraits	Introduction Identifying goals Introducing activities	PPT
		Activities Think about yourself and how others think of you Fold the A4 paper in two Express your self-portrait on the inside part and express others’ portraits of you on the outside part of the paper Attach the images from magazines by hand or with scissors using a collage method Draw additional pictures on the paper according to what you want to express	A4 paper MagazinesGlue & scissors WatercolorCrayon Marker
		Wrap-up Present their work Discussion about the participants’ artwork Q & A Notice for the next session	Presentation Discussion
XI (1 h)	Gift I want to receive, Gift I want to give to my family members	Introduction Identifying goals Introducing activities	PPT
		Activities Take time to think about the past Think about your family members and yourself Fold the A4 paper in two Write on the inside of the paper a gift you want to receive and on the outside of the paper write a gift you want to give to your family members Attach images torn from magazines using glue using a collage method	A4 paper MagazinesGlue WatercolorCrayon Marker
		Wrap-up Present their work Discussion about the participants’ artwork Q & A Notice for the next session	Presentation Discussion
XII (1 h)	Cooperative Salt “Mandala”	Introduction Identifying goals Introducing activities	PPT
		Activities Freely talk about your daily life with other group members Group members draw a “mandala” on the A2 paper Make colored salt by grinding pastels and mixing them into salt Complete the “mandala” with group cooperation using colored salt and sharing feelings	Pastel Salt A2 paper Glue
		Wrap-up Present their work Discussion about the participants’ artwork Sharing their experiences about GAT	Presentation Discussion

**Table 2 ijerph-18-05000-t002:** Results of the homogeneity test on the two groups’ demographic characteristics and outcome variables (*n* = 41).

Characteristics	Experimental (*n* = 20)		Control (*n* = 21)		χ^2^/t	*p*
	* N *	%	* N *	%		
Caregivers						
Gender						
Male	6	30	6	28.6	0.010	0.92
Female	14	70	15	71.4		
Age						
20–29	0	0	2	9.5	11.679	0.07
30–39	3	15.0	0	0.0		
40–49	0	0	5	23.8		
50–59	7	35.0	6	28.6		
60–69	8	40.0	6	28.6		
70–79	1	5.0	2	9.5		
80–89	1	5.0	0	0		
Religion						
Religious	13	65.0	14	66.7	1.178	0.882
Nonreligious	7	35.0	7	33.3		
Relation						
Spouse	11	55	8	38.1	4.785	0.31
Son	5	25	7	33.3		
Daughter	1	5	3	14.3		
Other	3	15	3	14.3		
Patient						
Gender						
Male	5	25.0	15	71.4	0.067	0.796
Female	15	75.0	6	28.6		
Age						
20–29	1	5	1	4.8	6.046	0.418
30–39	2	10.0				
40–49	1	5.0	3	14.3		
50–59	6	30.0	2	9.5		
60–69	4	20.0	6	28.6		
70–79	5	25.0	7	33.3		
80–89	1	5	2	9.5		
Diagnosis						
Injury by accident	2	10.0	3	14.3	1.253	0.74
Cerebral hemorrhage	7	35.0	6	28.6		
Cerebral infarction	11	55.0	12	57.1		
Duration						
<6 months	7	35.0	12	57.1	2.02	0.155
>6 months	13	65.0	9	42.9		
Number of hospitalizations						
1 time	5	25.0	2	9.5	3.949	0.413
2 times	6	30.0	8	38.1		
3 times	3	15.0	6	28.6		
4 times			1	4.8		
More than 5 times	6	30.0	4	19.0		
** Main variables **						
Depression	16.90 ± 10.13		18.71 ± 8.42		0.625	0.536
Burden	78.55 ± 9.67		73.24 ± 11.58		−1.579	0.119
Self-efficacy	74.95 ± 10.84		76.57 ± 8.26		5.041	0.592

**Table 3 ijerph-18-05000-t003:** Effects of the group art therapy intervention on depression, burden, and self-efficacy (*n* = 41).

Variables	Groups	Pre-Test	Post-Test	Paired-t (*p*)	*t*	*d*	*p*
		Mean (SD)	Mean (SD)				
Depression	Exp.	16.90 (10.13)	11.80 (7.35)	3.296 (0.004)	0.861	1.62	0.395
	Con.	18.71 (8.42)	16.14 (9.77)	1.046 (0.308)			
Burden	Exp.	78.55 (9.67)	71.15 (11.60)	4.564 (<0.001)	2.462	1.19	0.020
	Con.	73.24 (11.58)	74.81 (11.03)	−0.481 (0.635)			
Self-efficacy	Exp.	74.95 (10.84)	80.95 (8.86)	−5.395 (<0.001)	−6.270	1.38	<0.001
	Con.	76.57 (8.26)	74.19 (8.52)	3.211 (0.004)			

Note. Exp.: experimental group, Con.: control group, *d*: effect size (ES).

## Data Availability

Not Applicable.

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
