# Peer review of "The Effects of Group Art Therapy on the Primary Family Caregivers of Hospitalized Patients with Brain Injuries in South Korea"

_ijerph, 2021, doi:10.3390/ijerph18095000_

Round 1

Reviewer 1 Report

IJERPH review

BPW 4/23/2021

The Effects of Group Art Therapy on the Primary Family Care- 2

givers of Hospitalized Patients with Brain Injuries in South Korea

Dear editors:

Thank you for the opportunity to review this manuscript. The topic is of great interest to persons working in the field of brain injury, and especially relevant to persons who work with family members of those who have sustained a brain injury.

The research design is well-planned and analyses are carefully described. My comments are few, followed by suggestions to strengthen  clarity and highlight small grammatical errors.

Overall, the paper is well-written and offers important information to several professional fields working in this area.

Overall comments:

  1. I appreciate the careful wording in the Discussion section, particularly on line 411 where it state GAT is “complimentary to” nursing. The intervention is founded within art therapy or occupational therapy practice domains in many countries outside of S. Korea. Perhaps describing GAT in this study as collaborative practice between nursing and Art Therapy would serve well in respecting professional scopes of practice. Indeed, collaborative practice on rehabilitation teams is exemplary practice, but as written it implies that while the group consulted Art Therapy in the design of the intervention, it is described solely as a nursing intervention (line 399).

  1. There is no discussion regarding the possible potency of group support and communication with the other members in affecting the results. It may be that the power of the intervention was the group dynamic and mutual support derived from participating and not specifically the GAT itself. This confound may be impossible to disentangle from the activity itself. However, in the discussion the possibility of mutual support should be voiced, and appreciated as an element of the GAT intervention.

  1. In Discussion, beginning in line 351, about the ns finding in depression. I suggest adding description of the observed change in the right direction for the GAT group. It appears that there was possible meaningful change here but that the sensitivity of the measure combined with the complexity of depression as you noted, made significance difficult to obtain. In this case, a larger group sample to improve power and perhaps adding another measure of depressive symptoms would show that in fact, the GAT does affect depressive mood.

Minor comments:

 Line 69: As a stand-alone paper, it would be helpful to clarify the prior research “median value” of what assessments?

Line 75-76: Incomplete sentence.

Line 93: Change to “increasing” self-efficacy (parallel form)

Line 133: Not sure what this means? How did you try to alter environment?

Author Response

Dear Reviewer 1,

Thank you for your invaluable time.

We, authors checked and revised according to your comments with several times in-depth discussion.

Thank you again.

Warm Regards,

I submit the Revised manuscript and Author response to the Reviewers.

Reviewer 2 Report

Thoughtful manuscript, with important implications for caregivers faced with great challenges. Study design is explained well and randomization is a strength. Selection of instruments for pre- and post- comparison across the three measures is appropriate and also a strength. Details for the GAT program, with examples, provide critical context, particularly for future of generalizing these outcomes or expanding the program scope.

Notes with line items and suggestions are attached. Will highlight two considerations for improved analysis of GAT efficacy:

Highly suggest a paired t-test or Wilcoxon for these data. I'd like to see how perceptions and characteristics changed across time within individual, and then compare degree of change across the interventions.

Do you have sufficient power to consider hospital stay, burden of disease (patient characteristics) or age, sex, [other variable of interest] of the caregiver, in evaluating change in depression, burden, self-efficacy from the GAT? (Group by variable of interest and compare across Exp. and Con.)

I wonder if you have qualitative data / feedback on the program, which may be useful to include. A text analysis would not be required in this manuscript, but could add value. Well done.

Author Response

Dear Reviewer 2,

Thank you for your invaluable time.

We authors checked and revised with in- depth discussion.

We upload two files

  1) Revised Manuscript  and

  2) Author response to Reviewers.

Warm Regards,
